# Intrareticular charge transfer regulated electrochemiluminescence of donor–acceptor covalent organic frameworks

Rengan Luo [1,4], Haifeng Lv [2,4], Qiaobo Liao[3], Ningning Wang[1], Jiarui Yang[1], Yang Li[1], Kai Xi [3✉], Xiaojun Wu [2✉], Huangxian Ju [1] & Jianping Lei [1✉]

The control of charge transfer between radical anions and cations is a promising way for decoding the emission mechanism in electrochemiluminescence (ECL) systems. Herein, a type of donor-acceptor (D-A) covalent organic framework (COF) with triphenylamine and triazine units is designed as a highly efficient ECL emitter with tunable intrareticular charge transfer (IRCT). The D-A COF demonstrates 123 folds enhancement in ECL intensity compared with its benzene-based COF with small D-A contrast. Further, the COF's crystallinity- and protonation-modulated ECL behaviors confirm ECL dependence on intrareticular charge transfer between donor and acceptor units, which is rationalized by density functional theory. Significantly, dual-peaked ECL patterns of COFs are achieved through an IRCT mediated competitive oxidation mechanism: the coreactant-mediated oxidation at lower potential and the direct oxidation at higher potential. This work provides a new fundamental and approach to improve the ECL efficiency for designing next-generation ECL devices.

[1] State Key Laboratory of Analytical Chemistry for Life Science, School of Chemistry and Chemical Engineering, Nanjing University, Nanjing 210023, China. [2] Hefei National Laboratory for Physical Sciences at the Microscale, Synergetic Innovation of Quantum Information and Quantum Technology, CAS Center for Excellence in Nanoscience, and School of Chemistry and Materials Sciences, University of Science and Technology of China, Hefei 230026, China. [3] School of Chemistry and Chemical Engineering, Nanjing University, Nanjing 210023, China. [4] These authors contributed equally: Rengan Luo, Haifeng Lv. ✉email: xikai@nju.edu.cn; xjwu@ustc.edu.cn; jpl@nju.edu.cn

Electrochemiluminescence (ECL) is a light-emitting process involving excited state generation through an electrochemical reaction[1,2]. The ECL efficiency mainly depends on the charge transfer between the emitters and the coreactant/emitter[3,4]. In most cases, the emitters such as small molecules and nanocrystals are transformed into excited states through an intermolecular charge transfer process[5–9]. Alternatively, by integrating coreactants with emitters, the resultant systems can undergo intramolecular charge transfer to enhance ECL[10–12]. For example, an electron transfer between Au nanocluster emitters and the grafted N,N-diethylethylenediamine coreactants was accelerated to drastically enhance ECL[11]. In our previous work, a stepwise-oxidation induced intrareticular electron transfer based on 1,4-diazabicyclo[2.2.2]octane coordinated metal-organic framework was established for ECL enhancement[12]. These works improved ECL performance by shortening the distance between emitter and coreactant, thus an efficient charge transfer between the emitter and coreactant was realized. By improving the intramolecular charge transfer rate, the reticular structure provides a promising opportunity to design more efficient ECL emitters.

Different from metal-organic frameworks that metal nodes might quench the ECL, the covalent organic frameworks (COFs) are a class of metal-free crystalline porous materials and could be applied as ECL emitters[13]. Due to the predesignable nature of COFs[14,15], their charge transfer behaviors can be modulated by integrating functional building blocks into a long-range ordered covalent framework[16–18]. Thus, the formation process of excited COF species could be finely revealed during ECL generation, which provides a promising approach to decode the relationship between COF structure and ECL emission. Usually, the intrareticular charge transfer (IRCT) in COF structures can be realized via an interlayer and an intralayer migration[19]. Typically, the interlayer charge transfer process can be achieved by aligning π units into topologically ordered columnar π arrays[20–22]. Meanwhile, the intralayer charge transfer can be regulated by topology-templated conjugation and integrating donor and acceptor (D–A) units into a reticular skeleton[23–26]. Therefore, modulating the chemical structure of COFs is a promising way to realize efficient IRCT. For example, the sensitivity of stimuli-responsive solvatochromic COFs was promoted by tuning the aldehyde counterpart with different electron affinities, in which stronger charge transfer transitions were achieved in larger D–A contrasts[27]. D–A COFs have the ordered frameworks to stabilize electrons/holes obtained from electrode and coreactants[28], thus enabling the generation of excited states for ECL transduction through the IRCT process.

Taking advantages of D–A pairs, a luminescent COF was designed as an ECL emitter by integrating triazine and triphenylamine as donor and acceptor units in the reticular structure, respectively (Fig. 1a). The resulting D–A COF exhibited 123-fold ECL enhancement owing to the rapid IRCT in aqueous solution compared with its benzene-based COF counterpart. The COF's crystallinity- and protonation-modulated ECL behaviors further identified ECL dependence on IRCT, which was theoretically confirmed by the density functional theory. Significantly, the dual ECL peaks of COFs were realized through the coreactant-mediated oxidation mechanism along with the direct oxidation mechanism at different potentials. Overall, the D–A COFs not only provide a delicate approach to design efficient ECL emitters but also enrich the fundamentals of ECL signal transduction in optoelectronic devices.

## Results

### COF synthesis and characterization.
We delicately designed a series of tris(4-formylphenyl)amine (TFPA)-based COFs that are synthesized with 2,4,6-tris(4-aminophenyl)-1,3,5-triazine (TAPT),

1,3,5-tris(4-aminophenyl)benzene (TAPB), and tris(4-aminophenyl) amine (TAPA) counterparts, denoting as t-COF, b-COF, and a-COF, respectively (see the Methods section)[29]. Figure 1b, c shows the simulated eclipsed stacking structure of the hexagonal t-COF. Powder X-ray diffraction analysis of t-COF revealed a crystalline structure with diffraction peaks at $2\theta = 4.4, 7.7, 8.9, 11.8$, and $22.5°$, assigned to the 100, 110, 200, 210, and 001 facets, respectively (Fig. 1d). Powley refinements provide good fits to the experimental powder X-ray diffraction (PXRD) data, identifying the crystalline structures of three COFs (Fig. 1d and Supplementary Fig. 1). According to the cross-polarization magic-angle-spinning solid-state NMR experiment (Fig. 1e), the resonances at ~161 ppm, ~159 ppm, and ~158 ppm indicate the presence of imine bonds in t-COF, b-COF, and a-COF. The downfield shift of resonance for C=N in t-COF was attributed to the strong electron-withdrawing ability of triazine units[30,31]. Fourier transform infrared spectra of t-COF, b-COF, and a-COF showed vibration peaks at ~1620 cm⁻¹, confirming the imine-linked framework structures (Supplementary Figs. 2–4). Furthermore, we measured the nitrogen adsorption isotherms of these three COFs at 77 K. The Brunauer–Emmett–Teller surface areas were calculated to be $1372 \text{ m}^2 \text{ g}^{-1}$, $1689 \text{ m}^2 \text{ g}^{-1}$, and $1364 \text{ m}^2 \text{ g}^{-1}$ for t-COF, b-COF, and a-COF, respectively (Fig. 1f), identifying the porous structures. High-resolution transmission electron microscopy further confirmed the formation of honeycomb-like porous structures with periodicities of $1.7 \pm 0.2$ nm for t-COF (Fig. 1g and Supplementary Fig. 5) and $1.7 \pm 0.2$ nm for b-COF (Supplementary Fig. 6), and layered morphology of a-COF (Supplementary Fig. 7). Overall, we have successfully synthesized three TFPA-based COFs with highly crystalline and porous structures.

The bands gaps of t-COF, b-COF, and a-COF calculated from solid-state UV–Vis spectra (Supplementary Figs. 8,9) are 2.42 eV, 2.53 eV, and 2.28 eV, which are consistent with theoretical band gaps (Supplementary Table 1). Coupling with ultraviolet photoelectron spectroscopy (Supplementary Fig. 10), the energy levels of t-COF, b-COF, and a-COF were calculated (Fig. 2b), identifying the semiconducting structure of these COFs. With the screened hybrid HSE06 functional, the calculated electronic band structures of t-COF, b-COF, and a-COF along the symmetry line in the Brillouin zone from Γ to M and K were demonstrated in Supplementary Fig. 11. Distinct from graphene, flat bands exist near the Fermi energy level, further confirming that all three COFs are semiconductors. Moreover, benefiting from the conjugated D–A skeleton with non-planar building blocks, the photoluminescence (PL) intensity of t-COF at 607 nm is about ten folds stronger than that of a-COF in aqueous media (Fig. 2a and Supplementary Fig. 12)[32,33]. The highly luminescent t-COF semiconductor provides a promising candidate as ECL emitter in an aqueous solution.

### ECL properties of D–A COFs.
The ECL behaviors of t-COF were investigated in different conditions. As shown in Supplementary Figs. 13−16, t-COF as stable reticular crystalline material demonstrated efficient performance by using 1,4-dioxane and TPrA as dispersion solvent and co-reactant during anodic ECL measurements, respectively. Compared to b-COF, t-COF showed about 123 folds enhancement of ECL while a-COF showed almost no ECL (Fig. 2c), inferring that the D–A structure of t-COF plays vital role in ECL generation[34,35].

In contrast with b-COF and a-COF, the highest occupied state (HOS) of t-COF is predominantly located at triphenylamine donor units, and the lowest unoccupied state (LUS) of three COFs are delocalized over the framework, confirming the π-conjugated structures (Fig. 2d–f). The localization of HOS confirmed the formation of the largest D-A contrast structure of t-COF. In fact, the cathodic peaks of t-COF, b-COF, and a-COF at $-1.18$ V, $-1.15$ V,

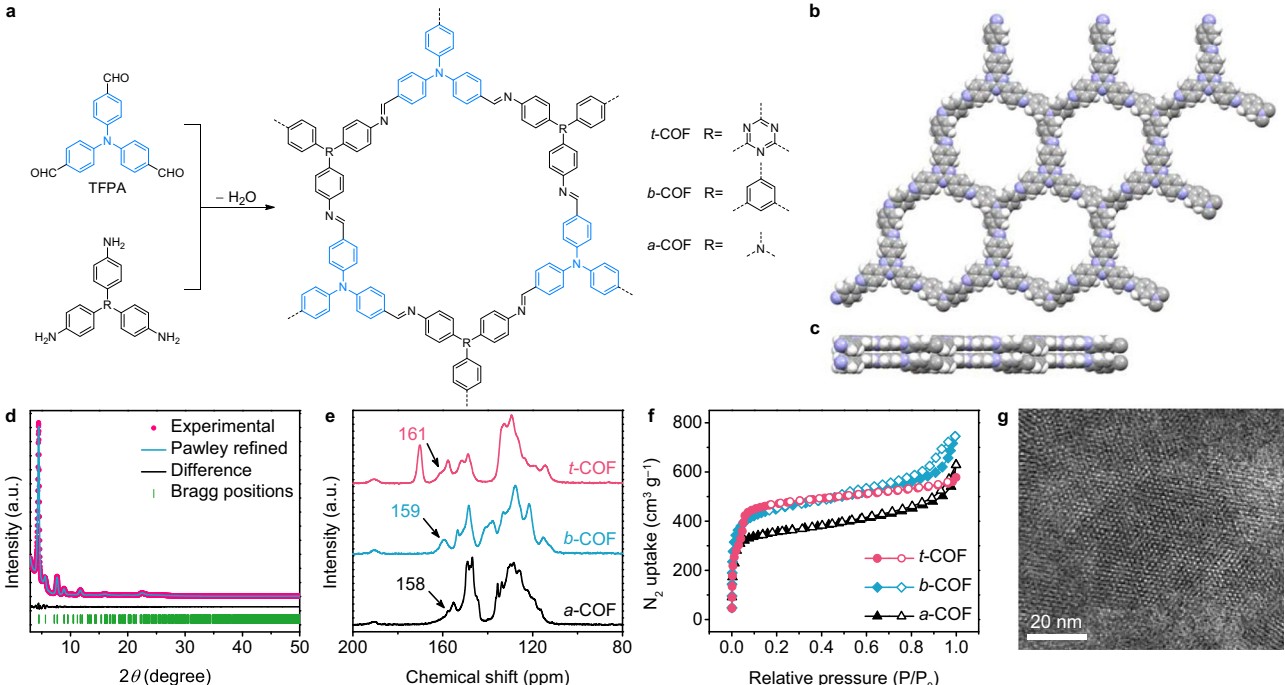

**Fig. 1 Synthesis and characterization of COF-based ECL emitters. a** Schematic illustration of synthesis of three TFPA-based COFs. Reconstructed crystal structure of *t*-COF: **b** top and **c** side views. **d** Experimental and simulated PXRD patterns of *t*-COF. **e** Solid-state ¹³C CP-MAS spectra and **f** N₂ sorption isotherms of *t*-COF, *b*-COF, and *a*-COF. **g** HR-TEM image of *t*-COF. Unweighted-profile R factor (Rp) = 1.53% and weighted-profile R factor (Rwp) = 2.20%.

and −1.14 V proved that the COFs are able to obtain electrons (Supplementary Fig. 17b). And the oxidation peaks of *t*-COF, *b*-COF, and *a*-COF appeared at +1.15 V, +1.05 V, and +0.85 V (Supplementary Fig. 17a,c), which contributed to the electron-deficient triazine units in the conjugated framework[36].

For comparison, we synthesized two model compounds, in which TAPT and TFPA were terminated by benzaldehyde and aniline, denoted as MC-TAPT and MC-TFPA, respectively. Interestingly, *t*-COF showed about nine folds ECL enhancement relative to MC-TFPA, while MC-TAPT showed nearly no ECL signal (Fig. 3a), indicating that the triphenylamine unit is essential for ECL emission and could serve as emission center. Furthermore, compared with the PL spectra of TFPA, TAPT, MC-TFPA, and MC-TAPT (Fig. 2a and Supplementary Fig. 18), the PL red-shift of *t*-COF also verified the conjugated D–A structure, providing a robust fundamental for enhanced ECL. In fact, the oxidation peaks of TFPA, MC-TFPA, and *t*-COF appeared at +1.49 V, +1.39 V, and +1.15 V (Supplementary Figs. 19,20), which was contributed to that the D–A conjugation lowered down the oxidation potential of triphenylamine unit[36].

In addition, ECL transients of *t*-COF obtained by stepping pulse were applied to investigate the intrareticular charge transfer between triazine and triphenylamine units (Supplementary Fig. 21). The alternation of potential was pulsed between +1.2 V and −1.3 V, which are sufficient to oxidize and reduce *t*-COF, respectively. Both anodic and cathodic steps demonstrated ECL emissions, which were attributed to the annihilation process between the triazine radical anions and triphenylamine radical cations through intrareticular charge transfer[37].

**IRCT-regulated ECL of *t*-COF.** The intrareticular charge transfer between two building blocks is supported by the time-correlated single-photon counting traces (Fig. 3b and Supplementary Fig. 22). The red-shifted absorption (Supplementary Fig. 23) and the reduced PL lifetime (Supplementary Table 2) of *t*-COF indicate the charge transfer process between electron-rich triphenylamine and electron-deficient triazine subunits[38]. To further verify the IRCT process of *t*-COF, a series of *t*-COFs were synthesized with different initial ratios of 1,4-dioxane and mesitylene (6:4, 5:5, 4:6, 8:2, and 2:8), in which the crystallinity was indicated by PXRD intensities (Fig. 3d). Interestingly, *t*-COF with higher crystallinity exhibited stronger ECL signal (Fig. 3c and Supplementary Fig. 24). The red-shift of PL peaks and reduced PL lifetimes of *t*-COFs were also consistent with the degree of crystallinity (Supplementary Figs. 25,26 and Supplementary Table 2)[39], which further confirmed that highly ordered conjugated skeleton of *t*-COF could facilitate charge transfer during the ECL process. The stable ECL signal under continuous cyclic voltammetry (CV) exhibited the good repeatability of IRCT during ECL generation (Supplementary Fig. 27).

Considering the IRCT between the donor and acceptor, protonation of imine bonds could slow down the charge transfer mobility and regulate the ECL behavior. Interestingly, with the increase of pH from 6.0 to 8.0, dual ECL peaks were observed at around +0.92 V and +1.05 V, which were denoted as ECL I and II, respectively (Fig. 4a and Supplementary Figs. 28,29). The peak potentials of *t*-COF oxidation were constant in the pH range, while the oxidation potentials of TPrA negatively shifted from +1.12 V to +0.98 V (Fig. 4b and Supplementary Fig. 29a), indicating the dependence of ECL signal on pHs. The ECL spectra collected under different pHs all peaked at around 604 nm (Fig. 4c), indicating the same excited states were generated for both ECL I and II as PL verified (Supplementary Fig. 30). At acidic environment, the TPrA and triphenylamine units of *t*-COF were simultaneously oxidized at around +1.12 V on the electrode, generating *t*-COF⁺• and TPrA⁺•. The latter could deprotonate and generate TPrA• to reduce the framework to *t*-COF⁻•, thus giving the ECL emission via the annihilation between *t*-COF⁺• and *t*-COF⁻• through IRCT. On the other hand, at basic environment, the TPrA was firstly oxidized to TPrA⁺•, and produced ECL I peak through a coreactant-mediated oxidation mechanism where TPrA was used as both oxidant and reductant[40,41].

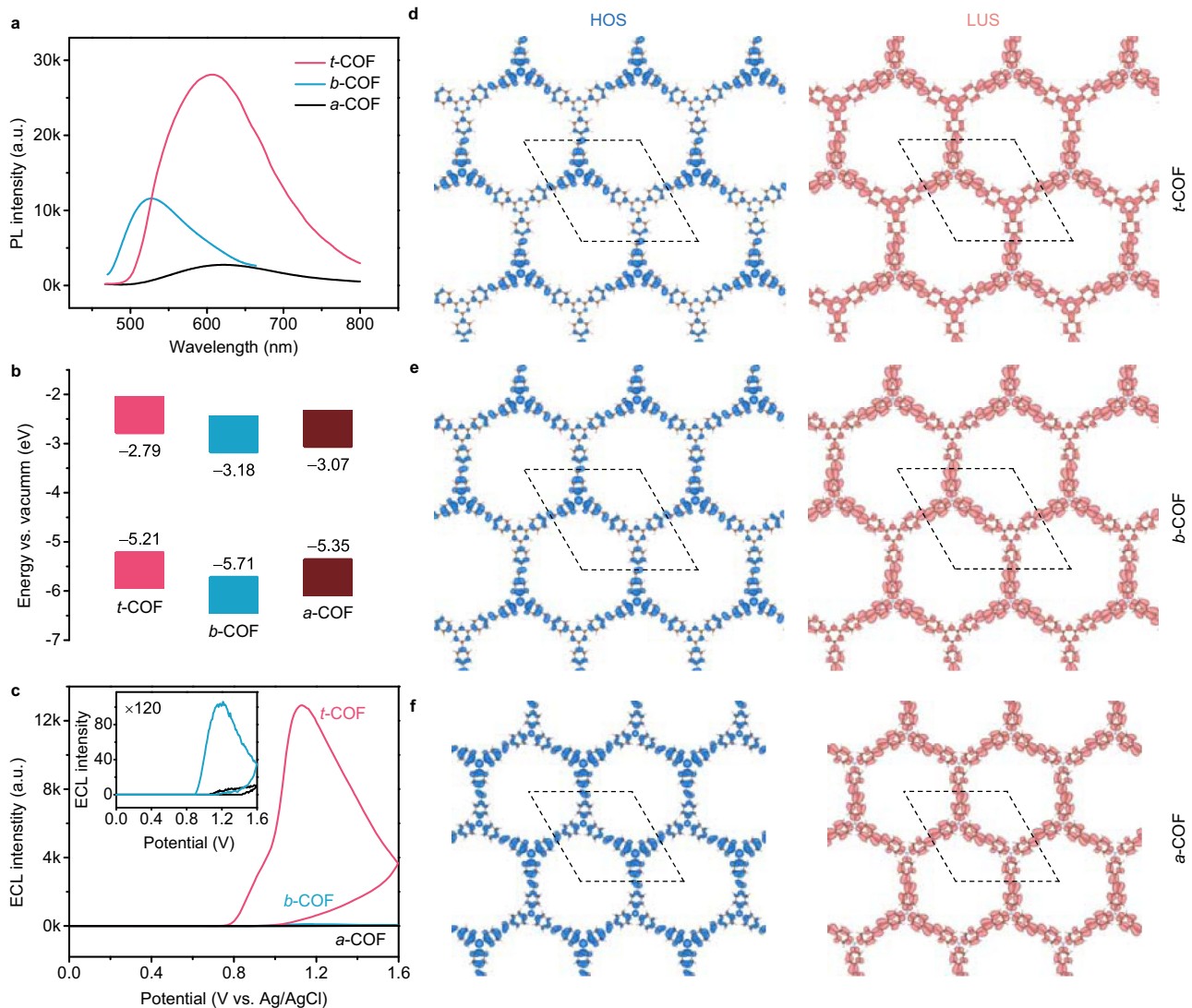

**Fig. 2 Spectral and ECL properties of D–A COF semiconductors. a** PL spectra and **b** energy levels of *t*-COF, *b*-COF, and *a*-COF. **c** ECL curves of three COFs modified GCEs in 0.10 M PBS (pH = 6.8) containing 20 mM TPrA (photomultiplier tube voltage = 700 V). Scan rate: 0.10 V s$^{-1}$. Inset: Magnified ECL curves of *b*-COF and *a*-COF. HOS and LUS of **d** *t*-COF, **e** *b*-COF, and **f** *a*-COF in top view. Isosurface is set to be 0.001 e bohr$^{-3}$.

Furthermore, the retained strength of the diffraction peaks of *t*-COFs at 4.4° indicated the stable crystalline structure in pH 6.0–8.0 (Supplementary Fig. 31). Meanwhile, the decrease of the relative ECL efficiency was observed with the decrease of pHs (Fig. 4d and Supplementary Fig. 32), which contributed to that the protonation inhibited the IRCT between triazine and triphenylamine units[25,42]. Indeed, dependent on protonation of imine bonds in *t*-COF structure, the effective mass of electrons increased from 0.23m$_0$ to 0.41m$_0$, while that of holes increased from 0.04m$_0$ to 0.16m$_0$ (Fig. 4e, f and Supplementary Table 3). According to the charge mobility equation (see the Methods section), the larger charge effective mass indicated a smaller charge carrier's mobility, thus a slower IRCT resulted in a lower ECL efficiency[43,44]. Overall, the dual ECL peaks of *t*-COF were realized through the crystalline and protonation-modulated IRCT process between D–A pairs.

**Competitive oxidation ECL mechanism**. As shown in Supplementary Fig. 33, the two oxidation peaks of CV curves of *t*-COF are well-matched with the two ECL peaks, inferring that ECL I is dependent on the oxidation of TPrA, while ECL II depends on the direct oxidation of *t*-COF at electrode surface. Furthermore,

the consistency of ECL spectroscopy (604 nm) with PL spectroscopy (607 nm) indicated the band-gap ECL model with the same excited states (Fig. 5a)[45]. When *t*-COF was oxidized, a new adsorption peak at 553 nm appeared due to the quinoid resonance structure (Supplementary Fig. 34a,35), which can be recovered by the following reduction under −1.2 V potential or cyclic voltammetry from +1.4 V to −1.4 V (Supplementary Fig. 34b)[46]. Also, the electron on natural transition orbital 152 was mainly distributed at the triazine part, suggesting the formation of quinoid species (Supplementary Figs. 36–38). Both UV–Vis spectra and theoretical calculation confirmed the transformation from aromatic to quinoid resonance structure. To describe the electron transfer process of reticular *t*-COF in electrochemical excitation, the 1st excited state is approximately simulated by moving one electron from the HOS to LUS. The electron density difference between 1st excited state and ground state demonstrated that the IRCT process involves electron density loss on the triazine units and electron density gain of the triphenylamine units, which is coherent with the charge distribution change of *t*-COF cell fragment and verifies the charge transfer between triphenylamine and triazine units (Fig. 5b and Supplementary Figs. 39–41)[47,48].

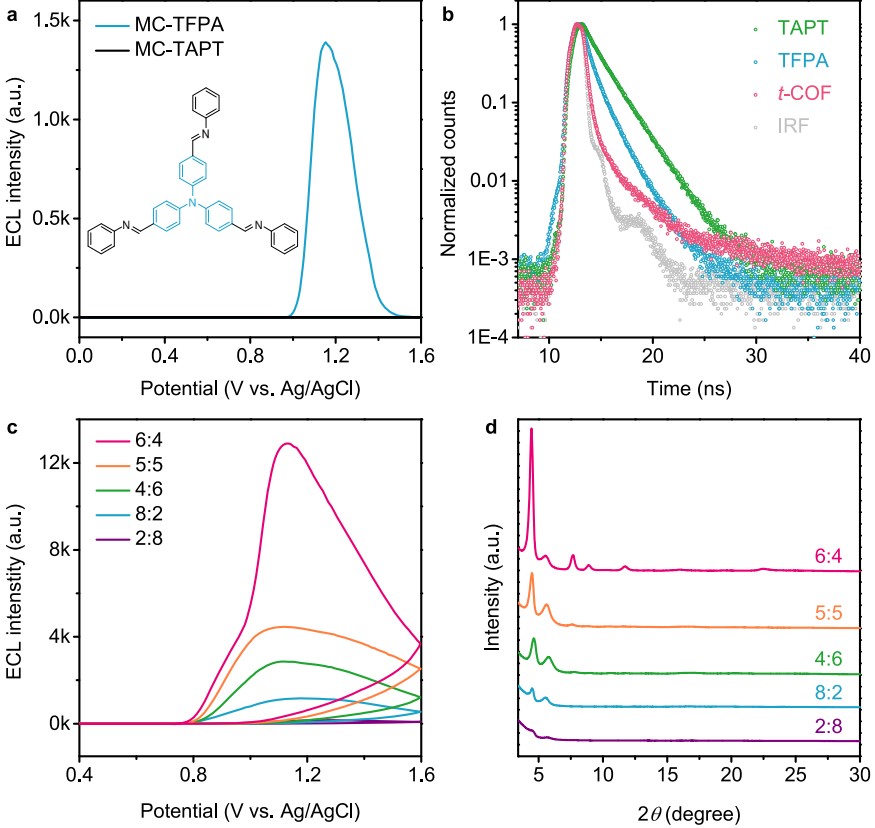

**Fig. 3 Crystallinity-modulated ECL of *t*-COF. a** ECL patterns of MC-TFPA and MC-TAPT in 0.10 M PBS (pH = 6.8) containing 20 mM TPrA. Inset: Chemical structure of MC-TFPA. **b** TCSPC traces of TAPT, TFPA, and *t*-COF. **c** Dependence of ECL on crystallinity and **d** corresponding PXRD spectra of *t*-COFs synthesized with the different ratios of 1,4-dioxane and mesitylene of 6:4, 5:5, 4:6, 8:2, and 2:8 (v:v) of *t*-COFs.

Theoretical calculations of the structure stability with different number of electrons and holes doping were conducted. When each cell was doped with one electron or hole, the lattice parameters of electron- and hole-doped *t*-COF ($a = 23.2285$ and $23.1064$ Å) are consistent with that of the pristine *t*-COF (Supplementary Fig. 42 and Supplementary Table 4), suggesting the good stability of the charged structure. When electrons were injected into the *t*-COF system (Fig. 5c), the radical anions at triazine units of *t*-COF are relatively stable compared to radical cations at triphenylamine units. Furthermore, density of states patterns exhibited that HOS/LUS moved to the Fermi level when holes/electrons were doped (Fig. 5d), and therefore enabled the efficient charge transfer.

In a word, the dual ECL peaks were attributed to two mechanisms. That is, the triazine unit of *t*-COF gained electrons from the resulted TPrA$^\bullet$ to generate *t*-COF$^{-\bullet}$ while *t*-COF$^{+\bullet}$ was obtained from the triphenylamine unit oxidized by either TPrA$^{+\bullet}$ (Fig. 5e, left) or electrode (Fig. 5e, right). Finally, upon annihilation between triazine radical anions and triphenylamine radical cations through the intrareticular charge transfer, dual ECL peaks were realized via a competitive oxidation mechanism: coreactant-mediated oxidation at lower potential and direct oxidation at higher potential, providing a new proof of concept for ECL fundamentals.

## Discussion

By coupling the electron-rich TFPA as electron donor with different electron-deficient amino counterparts as electron acceptors, we designed three kinds of COF-based ECL emitters for the first time. As the electron affinities of amino units are in an order of: triazine > benzene > triphenylamine, the D-A contrasts of constructed COFs are in the order of: *t*-COF > *b*-COF > *a*-COF. Moreover, large D-A contrast leads to localization of HOS with high conjugation in the

framework, which is expected to produce efficient charge separations. Also, the low relative energy of electrons-doped *t*-COF verified the stable radical anions of *t*-COF. Compared to *b*-COF, *t*-COF demonstrated about 123 folds enhancement of ECL while *a*-COF showed no obvious ECL (Fig. 2c). So, the ECL performances of these COFs are in the order of *t*-COF > *b*-COF > *a*-COF.

Since highly crystalline COFs have ordered conjugated skeleton, the efficient charge transfers are likely conducted among triazine and triphenylamine units. The crystalline *t*-COFs could efficiently boost IRCT between donors and acceptors, and stabilize radical cations and anions, thus significantly enhancing the ECL performance. On the other hand, when the imine bonds in *t*-COF were protonated, the effective masses of electrons increased from 0.23 $m_0$ to 0.41 $m_0$, suggesting the attenuated mobility of the charge carriers. Therefore, ECL efficiencies of *t*-COF decreased with the decrease of pHs (Fig. 4d). We proposed that the efficient ECL of *t*-COF is attributed to the stabilized radical triazine anions and rapid IRCT of the conjugated D-A structure.

The intriguing dual-peaked ECL was attributed to different oxidation pathways of triphenylamine units. Varying pH from 6.0 to 8.0, the oxidation potentials of TPrA negatively shifted from +1.12 V to +0.98 V while that of *t*-COF was constant at around +1.15 V. At basic environment, it turns out that the TPrA is oxidized earlier than *t*-COF, thus providing interval for TPrA$^{+\bullet}$ to react with *t*-COF through coreactant-mediated oxidation mechanism, resulting in ECL I. At acidic environment, TPrA and *t*-COF are oxidized at the electrode surface simultaneously, direct oxidation mechanism takes place and ECL II has appeared. Based on the competitive oxidation of triphenylamine units, dual ECL peaks were realized via coreactant-mediated oxidation mechanism at lower potential and direct oxidation mechanism at higher potential.

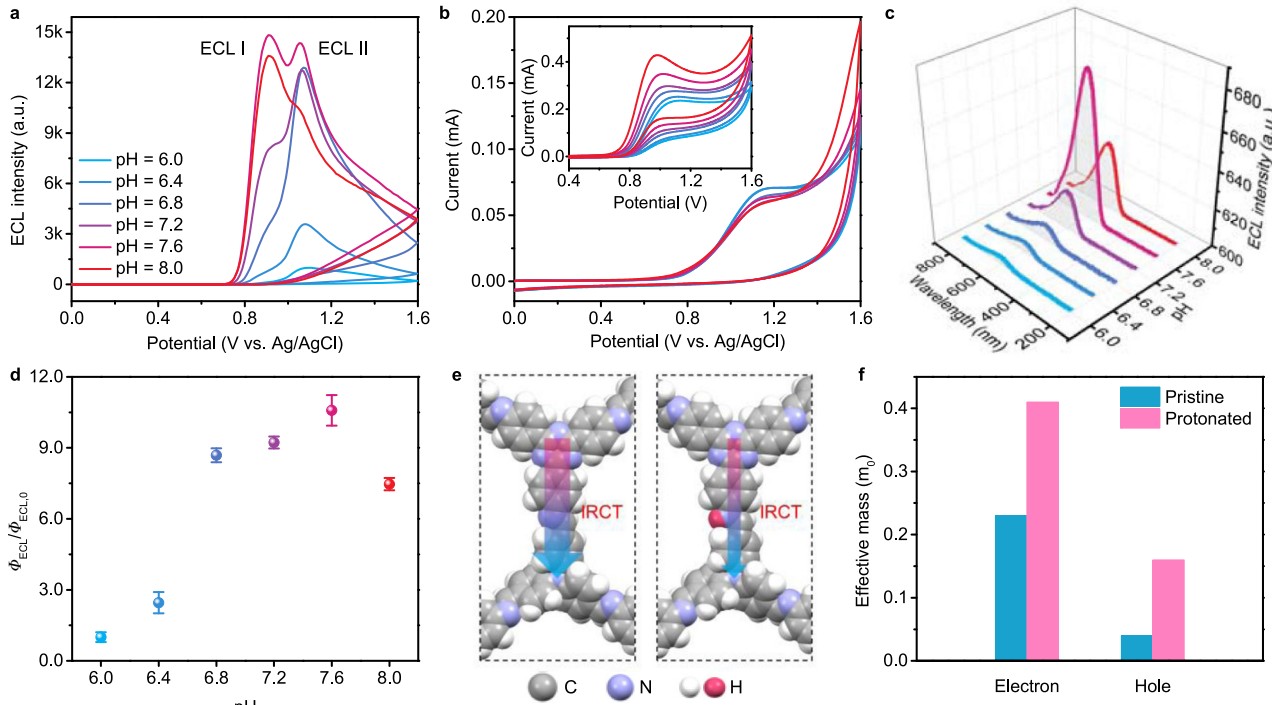

**Fig. 4 Protonation-modulated dual ECL peaks of *t*-COF. a** Dual-peaked ECL patterns containing 20 mM TPrA and **b** CV curves of *t*-COFs modified GCEs in 0.10 M PBS at different pHs. Inset: CV curves of TPrA on a bare GCE at different pHs. **c** ECL spectra of *t*-COF collected in 0.10 M PBS containing 100 mM TPrA at different pHs. **d** Dependence of the relative ECL efficiency ($\Phi_{ECL}/\Phi_{ECL,0}$) on pHs (the error bars represent the s.d. from triplicate measurements). $\Phi_{ECL}$ and $\Phi_{ECL,0}$ represent the ECL efficiencies of *t*-COF in given pHs and pH 6.0, respectively. **e** Structure and **f** effective electron and hole mass of pristine and protonated *t*-COF.

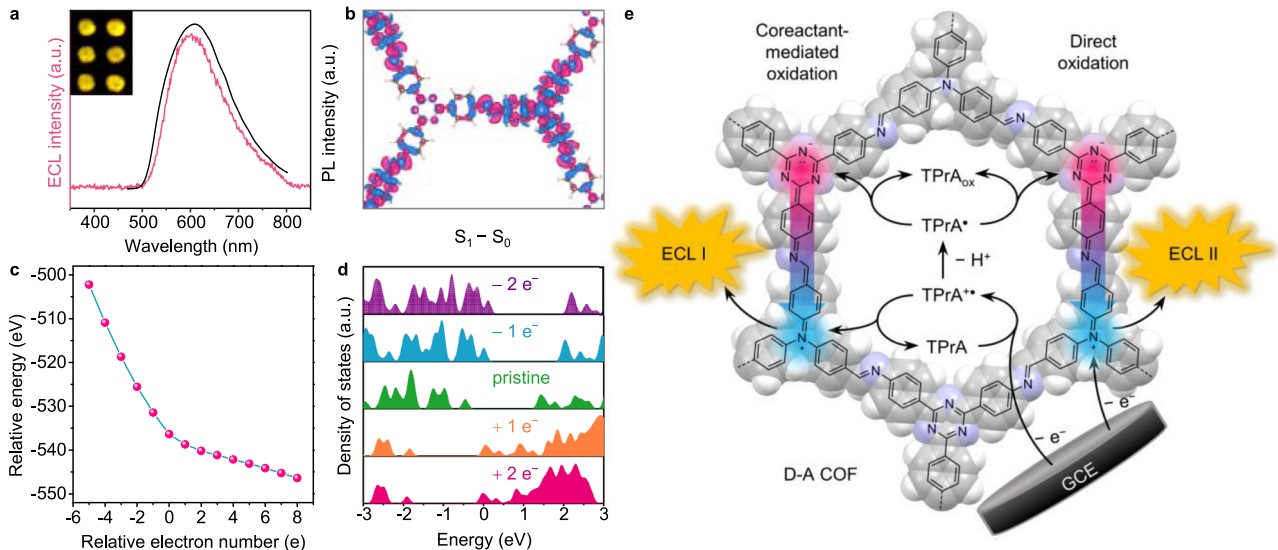

**Fig. 5 Competitive oxidation ECL mechanism of *t*-COF. a** ECL and PL spectra of *t*-COF. Inset: ECL images of *t*-COF at the applied potential of +1.40 V in 100 mM TPrA solution. **b** The difference of *t*-COF charge density between 1st excited state and ground state. Electron density gain and loss are donated as blue and red, respectively. **c** Relative energy and **d** density of states of *t*-COF doped with different electron numbers. **e** Schematic illustration of ECL I and II generation mechanism via intrareticular charge transfer.

In summary, to regulate the charge transfer between radical anions and cations in ECL generation, a TFPA-based COF as an efficient ECL emitter was designed by tuning the amino counterparts with different electron affinities. Due to the stable radical triazine anions and rapid IRCT, highly crystalline *t*-COF showed about 123 folds enhancement of ECL compared to the benzene-based *b*-COF. The attenuated mobility of electrons mediated by the protonation of the imine linkers further identified the dependence of ECL generation on IRCT, which was theoretically rationalized by DFT. Moreover, electron density difference calculation between the first excitation and ground state further proved the IRCT between triazine and triphenylamine. More significantly, dual ECL peaks were achieved via the coreactant-mediated oxidation mechanism and the direct oxidation

mechanism. Overall, the ordered D-A structure of COF not only provides a new avenue to design ECL emitter for enhanced ECL but also gives an intrareticular charge transfer pathway to decode the ECL essential.

## Methods

**Synthesis of *t*-COF, *b*-COF, and *a*-COF.** The synthesis method of *t*-COF follows previously reported procedures[29]. TFPA (32.9 mg, 0.010 mmol) and TAPT (35.4 mg, 0.010 mmol) are weighed into a 15 mL glass tube. 2 mL of 1,4-dioxane:mesitylene (v:v = 6:4) and 0.2 mL of acetic acid (6 M) solution were added to the tube, which was sonicated thoroughly. The tube was degassed by the freeze–pump–thaw technique three times to achieve an internal pressure of ~50 mTorr and flame sealed under vacuum. The reaction was heated at 120 °C for 84 h to yield an orange solid at the bottom of the tube. The wet sample was filtrated and transferred to a Soxhlet extractor and thoroughly washed with THF and methanol for 24 h. Finally, the solid was subjected to supercritical CO₂ drying to produce *t*-COF (yield: 95%). *t*-COFs with different degrees of crystallinity were synthesized using different volume ratios of 1,4-dioxane and mesitylene (5:5, 4:6, 8:2, and 2:8). The synthesis of *b*-COF and *a*-COF followed similar procedures as *t*-COF, except that TAPT was replaced with TAPB (35.2 mg, 0.010 mmol) and TAPA (29.0 mg, 0.010 mmol), respectively.

**ECL measurement and efficiency calculation.** The electrochemiluminescence (ECL) measurements were conducted using a three-electrode system in a quartz electrolytic cell. The three electrodes include a 5 mm diameter glassy carbon electrode (GCE) as working electrode, a Pt counter electrode, and a Ag/AgCl (saturated) reference electrode, and ECL was carried out in 0.10 M PBS containing 0.10 M KNO₃ with 20 mM TPrA. COF powders were dispersed in 1,4-dioxane and sonicated for 2 h to form 1.0 mg mL⁻¹ COF/1,4-dioxane suspensions. 5 μL of the suspension was drop-coated on a clean GCE and dried at room temperature. The relative ECL efficiencies were evaluated by the ratios of $\Phi_{ECL}$, which can be calculated as below[49]:

$$\Phi_{ECL} = \frac{\int_0^t I dt}{\int_0^{t'} i_a dt'} = \frac{\int_0^{t'} I dt}{Q_a}$$

Where $I$ (einsteins s⁻¹) represents the total ECL intensity integrated over the testing period of time $t'$ (s) and $i_a$ represents integration of the anodic current over the same time period that results in the total anodic charge $Q_a$.

**Calculation parameters.** Non-spin-polarized first-principle calculations were carried out based on density functional theory (DFT) framework by utilizing the Vienna Ab initio Simulation Package 5.4.4 package[50,51]. Pseudopotential was used to describe the electron–ion interactions within the Projector Augmented Waves approach[52,53]. Generalized gradient approximation of Perdew-Burke-Ernzerhof was adopted for exchange-correlation potential[52–54]. To better describe the interlayer van der Waals interactions, we used the most widely used Grimme's D3 dispersion corrections for optimization of bulk *t*-COF, *b*-COF, and *a*-COF[55]. Band structure calculations based on DFT with local or gradient-corrected exchange-correlation potentials were known to severely underestimate the bandgap of semiconducting and insulating materials. To calculate the band structures and specify the energy levels of *t*-COF, *b*-COF, and *a*-COF, we adopted a more accurate hybrid functional Heyd–Scuseria–Ernzerhof (HSE06) that incorporates a portion of exact exchange from Hartree–Fock theory with the rest of the exchange–correlation energy from other sources (ab initio or empirical), which have been shown to give good results for most systems[56,57]. The electron wave functions were expanded on a plane-wave basis set with an energy cut-off of 500 eV. The atomic coordinates of all structures were allowed to relax until the forces acting on the ions were less than 0.01 eV Å⁻¹. Convergence criterion for the electronic self-consistent cycle is fixed at 1 × 10⁻⁵ eV. The integrations in the reduced Brillouin zone were performed on a Γ-centered 2 × 2 × 1 Monkhorst-Pack special k-points and a Γ-centered 4 × 4 × 1 Monkhorst-Pack special k-points for optimization and self-consistent calculations of two-dimensional (2D) *t*-COF, *b*-COF, and *a*-COF. For optimization and self-consistent calculations of bulk *t*-COF, *b*-COF, and *a*-COF, we adopted a 2 × 2 × 10 and a 4 × 4 × 12 Monkhorst-Pack special k-points[58,59]. For 2D *t*-COF, *b*-COF, and *a*-COF, a vacuum slab above 15 Å was used in all calculations to avoid the interlayer interactions. For the electron and hole-doped system, additional background charges are introduced to balance the electrical neutrality. For density of states calculations, we adopt a 6 × 6 × 14 Monkhorst-Pack special k-points.

**Calculation of effective mass for 2D *t*-COF.** According to the charge mobility formula of two-dimensional hexagonal lattice material:

$$\mu = \frac{eh^3 C_{2D}}{12\pi^3 k_B T |m^*|^2 E_1^2}$$

Where $\mu$ is the charge mobility, $e$ is the elementary charge, $h$ is the Planck constant, $C_{2D}$ is Young's modulus, $k_B$ is the Boltzmann constant, T can be taken as 300 K, $E_1$ is

the deformation potential, and $m^*$ is the effective mass. Before and after protonation, the mechanical properties of the material do not change significantly, so the mobility of charge transfer can be estimated by the effective mass. In each unit cell of pristine, the protonated 2D *t*-COF was constructed by adding one hydrogen atom bonded with the linker nitrogen atom. The protonated 2D *t*-COF consists of 42 carbon atoms, 28 hydrogen atoms, and 7 nitrogen atoms. The lattice parameters of protonated 2D *t*-COF were $a$ = 23.19 Å, $b$ = 23.23 Å; $\alpha = \beta$ = 90° and $\gamma$ = 119.89°. Here, we optimized the protonated 2D *t*-COF using the non-spin-polarized calculation and the bandgap is calculated to be 1.06 eV. The effective mass of pristine and protonated 2D *t*-COF is calculated along the direction of $\Gamma$ (0, 0, 0) to $M$ (0.5, 0, 0). The effective mass was calculated by fitting the quadratic differential coefficient of conduction band minimum (CBM) and valence band maximum (VBM).

**Charge density difference of 2D *t*-COF.** To describe the electron transfer process of 2D *t*-COF, the 1st excited state is approximately simulated by moving one electron from the highest occupied state (HOS) to lowest unoccupied state (LUS). The whole system of 2D *t*-COF owns total 230 valence electrons. Thus, the band indexes for HOS and LUS are 115 and 116, respectively. Then, we fix the occupancy of HOS and LUS through FERWE and FERDO parameters. Spin-up and spin-down orbitals are split to avoid the fractional occupancy. Subsequently, we move one electron (1e⁻) from HOS to LUS in the spin-up channel and for spin-down channel, the occupancy of electrons remains unchanged. Hence, the 1st excited state of *t*-COF could be regarded as moving one electron from the HOS to LUS. Based on the optimized 1st excited state of *t*-COF and pristine *t*-COF, we calculated the corresponding charge density in the real space. The charge density difference ($\rho_{diff}$) between 1st excited state and ground state is calculated using formula shown as below:

$$\rho_{diff} = \rho_{1st\ excited\ state} - \rho_{ground\ state}$$

## Data availability

The data that support the findings of this study are available within the article and supplementary information files, or from the corresponding author upon request.

## Code availability

The authors did not use any previously unreported custom computer code or algorithm.

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

## Acknowledgements
We gratefully acknowledge the National Natural Science Foundation of China (21675084, 22073087), the Natural Science Foundation of Jiangsu Province (BZ2021010), CAS Project for Young Scientists in Basic Research (YSBR-004), MOST (2018YFA0208603, 2016YFA0200602), and supports from Super Computer Centre of USTCSCC and SCCAS.

## Author contributions
R.G. Luo, H.F. Lv and J.P. Lei proposed the idea and designed the experiments. R.G. Luo, H.F. Lv, X.J. Wu and J.P. Lei wrote the manuscript. R.G. Luo carried out the synthesis and electrochemical experiments. H.F. Lv carried out the structure simulation and theoretical calculations. Q.B. Liao performed the sorption experiments. N.N. Wang helped in electrochemiluminescence measurements. J.R. Yang and Y. Li helped in COFs and model compounds synthesis. K. Xi, X.J. Wu, H.X. Ju and J.P. Lei supervised and coordinated all investigators for this project. All authors discussed the results and commented on the manuscript.

## Competing interests
The authors declare no competing interests.
