## [Peer Review File · Nature Communications]

REVIEWER COMMENTS

Reviewer #1 (Remarks to the Author):

In this manuscript, the authors have designed and synthesized three kinds of 2D COFs, which were used as ECL emitters. According to their electrochemiluminescence behavior, a COF with triphenylamine and triazine unit exhibited the best performance, due to the tunable intrarecticular charge transfer (IRCT). In addition, the authors found the electrochemiluminescence performance of these COFs strongly depends on their crystallinity and the protonation of the C=N bonds. Moreover, they also rationalized two mechanisms for the donor-acceptor COFs mediated with intrarecticular charge transfer. The work has been carried out well and the experimental/analytical characterizations are convincing. Indeed, the result is interesting. However, the authors should address the following questions before considering the publication:

1. How about the stability of these COFs under ECL experiments? If not, are they really COFs or just COFs nanosheets?
2. Usually, the protonation of C=N bonds will decrease the crystallinity of COFs. The author should reconsider this part.
3. What is the most significant advantage of COFs for ECL study? The authors should give more explanation in the introduction part.
4. The assignment of solid-C NMR is not good. The authors should perform the experiment again.
5. The authors collected ECL patterns of t-COF at different pHs from 6.0 to 8.0. Why do they choose these pHs? How about other pHs?

Reviewer #2 (Remarks to the Author):

In this work, Lei et al, presented the synthesis of a type of donor-acceptor covalent organic frameworks comprising triphenylamine and triazine units. These COFs were investigated as ECL emitter. The authors tried to establish some correlation between the COF structures and ECL behaviors. Unfortunately, some experimental results and phenomena could be clearly characterized and rationally explained, despite the authors added a large amount of theoretical modeling work. Some fundamental errors could be found. Therefore, we could not recommend such work suitable for submission in the present state. Some issues must be professionally checked and rationally resolved, the detailed was listed below:

1. The synthetic work is too common to attract the readers within a broad scope. Many similar structures have been reported and fully investigated, although this work tried to open up some special application of COFs.

2. The theoretical evaluation seems to occupy a high ratio in this work. As contrast, the experimental data are too less to provide solid supports for the discussion and results. In fact, a lot of characterization and analytic methods have been developed and widely used for the elucidation of semiconducting structure or properties of various organic materials, but not to be used in this work.
3. The authors emphasized on the charge transfer between triazine and triphenylamine units, however, the imine linkages between these acceptors and donors seem to seriously impede their communications via electron delocalization. Such interaction is not rational.
4. The stability of the resulting COFs against the different pH values had to be examined.
5. Dependence of ECL on crystallinity and corresponding PXRD spectra of 4t-COFs samples from the different ratios of 1,4-dioxane and mesitylene should be further carefully explored and interpreted. Some characterization including fluorescence lifetime measurement etc should be provided.
6. In supplementary Figure 13, the CV profiles look quite bad. It is difficult to find some defined information from the cathodic CV curves of the three COFs. The poor processibility for COF materials might make them not suitable for characterization via CV measurements. The quality of the COF-based electrode should be carefully explored either through processing method or optimizing the additives, e.g. conductive component.
7. We could agree with the presented ECL generation pathway of these COF. The resonance structures of aromatic and quinoid types (as shown in Figure 5 or Supplementary Figure 26) seems to be impossible. Either the triazine moiety or imine unit could not undergo such resonance structures due to the large negativity of nitrogen atoms in these units. While, there is no any evidence, such as from ESR or UV-vis-NIR for verifying the proposed mechanism.

Reviewer #3 (Remarks to the Author):

The authors present a novel method of improving the ECL efficiency by controlling the charge transfer processes. A donor-acceptor (D-A) covalent organic framework (COF) with triphenylamine and triazine units is designed as a highly efficient ECL emitter, which could be modulated by crystallinity and protonation. Dual-peaked ECL patterns of COFs are achieved through an IRCT mediated competitive oxidation mechanism: the coreactant-mediated oxidation at lower potential and the direct oxidation at higher potential. This is an interesting work that deserve publication after revision.

Comments:

1. The bands gaps of the three kinds of COFs were calculated by UV-vis spectra in Figure 2a. The values were 2.42 eV (~513 nm), 2.53 eV (~491 nm), and 2.28 eV (~545 nm) for t-COF, b-COF, and a-COF, respectively. However, the maximum photoluminescence intensities of the three kinds of COFs were observed at ~520 nm for b-COF and ~600 nm for t-COF and a-COF (Figure 2b). I wonder why the stokes shifts were so large. In addition, the bands gaps measured by UV-vis spectra in Figure 2a

were larger than the electrochemical results in Figure S13. The cathodic peaks of three COFs were at -1.15 V (Page 6, Line 107) and the anodic peaks of t-COF, b-COF, and a-COF were at 1.15 V, 1.05 V, and 0.85 V (Page 6, Line 109). As a result, the band gaps calculated by electrochemical results were 2.3 eV, 2.2 eV, and 2.0 eV for t-COF, b-COF, and a-COF.

2. As demonstrated by the authors, the ECL generation of t-COF is very efficient because of the largest donor-acceptor contrast structure of t-COF. However, the ECL performance of t-COF was only compared with other types of COFs. To further confirm the efficient ECL generation of t-COF, the ECL efficiency of t-COF, e.g., the relative ECL efficiency of t-COF with respect to Ru(bpy)₃²⁺, should be calculated.

3. In Figure 2b, the authors compared the photoluminescence intensities of t-COF and a-COF and the PL intensity of t-COF at 607 nm is about 10 times stronger than that of a-COF. However, the maximum PL intensities of the two COFs were different. The authors should explain why PL intensity at 607 nm is so important. Moreover, since COF is almost insoluble in all solvents, how to control the two kinds of COF are tested at the same concentration?

4. The strong π - π interactions and rotationally labile imine linkages make most of the solid state imine-linked COFs non-fluorescent due to fluorescence quenching processes (Chem. Commun 2018, 54, 2349-2352). For reader's interest, please explain why the three imine COFs in this work have strong fluorescence.

5. In Figure 5e, in basic environment, t-COF^{+•} was generated from the triphenylamine unit oxidized by either TPrA[•] or electrode. Meanwhile, the triazine unit of t-COF gained electrons from TPrA[•] and generate t-COF^{-•}. Therefore, there are two ECL signals generated from the annihilation between t-COF^{+•} and t-COF^{-•} and t-COF^{+•}, respectively. However, in acidic environment, there is only one ECL signal generated from the annihilation between t-COF^{+•} and t-COF^{-•}. Why different in basic and acid solutions?

6. In the course of electrode reaction, is t-COF consumed all the time or recycled like metal complexes, such as Ru(bpy)₃²⁺?

7. Figure S18 shows the CV curves of t-COF, "PMT = 700 V" in the caption is confusing, which might refer to PMT voltage.

Dr. Prof. Jianping Lei
State Key Laboratory of Analytical Chemistry for Life Science
School of Chemistry and Chemical Engineering
Nanjing University
Nanjing 210023, PR China
Email: jpl@nju.edu.cn

September 10, 2021

Re: “Competitive oxidation electrochemiluminescence mechanism of donor-acceptor covalent organic frameworks mediated with intrarecticular charge transfer” (NCOMMS-21-19890A)

Dear Sir,

Thank you very much for your valuable comments regarding our above paper. According to your opinions and suggestions, we have revised the manuscript. May I reply to your comments and show you the changes in the revision as follows:

Reviewer #1:

***Comment 1:** In this manuscript, the authors have designed and synthesized three kinds of 2D COFs, which were used as ECL emitters. According to their electrochemiluminescence behavior, a COF with triphenylamine and triazine unit exhibited the best performance, due to the tunable intrarecticular charge transfer (IRCT). In addition, the authors found the electrochemiluminescence performance of these COFs strongly depends on their crystallinity and the protonation of the C=N bonds. Moreover, they also rationalized two mechanisms for the donor-acceptor COFs mediated with intrarecticular charge transfer. The work has been carried out well and the experimental/analytical characterizations are convincing. Indeed, the result is interesting. However, the authors should address the following questions before considering the publication:*

Response: We are extremely grateful for the Respected Reviewer’s rating of the importance of the reported results. We have improved this work according to your insightful and constructive comments as follows:

***Comment 2:** How about the stability of these COFs under ECL experiments? If not, are they really COFs or just COFs nanosheets?*

Response: Thank you for your valuable comment. According to your suggestion, we have tested the stability of *t*-COF during ECL experiment. The PXRD patterns of *t*-COFs before and after electrochemical reaction showed no obvious changes (Fig. R1), suggesting the high stability of *t*-COF under ECL experiments. In addition, the constant ECL emissions under continuous CV scans also indicated the excellent stability of *t*-COF (Supplementary Fig. 27). We have supplemented Fig. R1 as Supplementary Fig. 16 in Supporting Information and the related explanation in page 6, line 5.

Figure R1. PXRD patterns of *t*-COF before and after ECL test in 0.10 M PBS (pH = 6.8) containing 20 mM TPrA.

Comment 3: Usually, the protonation of C=N bonds will decrease the crystallinity of COFs. The author should reconsider this part.

Response: To investigate the dependence of crystallinity of COFs on pHs, we exposed *t*-COF samples to 0.10 M PBS with different pHs for 12 h. The samples were then transferred to a Soxhlet extractor and thoroughly washed with THF and methanol for 24 h. Finally, the solids were subjected to supercritical CO₂ drying. We collected the PXRD patterns of these *t*-COF (Fig. R2). The strength of the diffraction peaks at 4.4° indicated that *t*-COFs retained their original skeleton and crystalline structure in pH 6.0–8.0. But at more acidic and basic environment, the crystallinity of *t*-COFs decreased due to hydrolysis. On the other hand, upon protonation of imine bonds in *t*-COF structure, the increased effective mass of electrons indicated a smaller charge carrier's mobility (Fig. 4e,f). Thus, the decrease of relative ECL efficiency in pH 6.0–8.0 was contributed to that the protonation inhibited the IRCT between triazine and triphenylamine units (Fig. 4d). We have supplemented Fig. R2 as Supplementary Fig. 31 in Supporting Information and the related explanation in page 8, last paragraph, lines 1–2.

Figure R2. PXRD patterns of *t*-COFs treated with 0.10 M PBS in pH 5.0–9.0 for 12 h.

Comment 4: What is the most significant advantage of COFs for ECL study? The authors should give more explanation in the introduction part.

Response: Thank you very much for your suggestion. Different from metal-organic frameworks that metal nodes might quench the ECL, the covalent organic frameworks (COFs) are a class of metal-free crystalline porous materials, leading to a highly crystalline ECL emitter. Due to the predesignable nature of COFs (*Chem. Rev.* **2020**, *120*, 8814), the most significant advantage of COFs for ECL study is that their charge migration in COF skeleton can be controlled by integrating building blocks with different functions, and thus the formation process of excited COF species could be finely revealed during ECL generation. Therefore, COFs as a suitable candidates provide the opportunities to decode the structure-ECL emission relationship in ECL study. We added the advantages of COFs as ECL emitters in page 3, last paragraph, lines 5–7.

Comment 5: *The assignment of solid-C NMR is not good. The authors should perform the experiment again.*

Response: According to your suggestion, we have performed the solid-C NMR again and assigned the carbon chemical shifts of *t*-COF, *b*-COF, and *a*-COF (Fig. R3). The signal to noise ratios of the spectra were obviously improved while the chemical shift at 161 ppm of *t*-COF, 159 ppm of *b*-COF, 158 ppm of *a*-COF verified the formation of imine bond. We updated Fig. 1e in revision.

Figure R3. Solid-state ^{13}C CP-MAS spectra of (a) *t*-COF, (b) *b*-COF, and (c) *a*-COF. Asterisks denote the spinning side bands.

Comment 6: *The authors collected ECL patterns of t-COF at different pHs from 6.0 to 8.0. Why do they choose these pHs? How about other pHs?*

Response: At pH from 6.0 to 8.0, the crystalline structure of *t*-COF was relatively retained as shown in Fig. R2, but the crystallinity of *t*-COF decreased at more acidic and basic environment. Thus, we collected ECL patterns of *t*-COF at different pHs from 6.0 to 8.0. Also, we investigated the ECL patterns at pH 4.0, 5.0, 9.0, and 10.0 (Fig. R4). All ECL patterns demonstrated single peaks at around +1.05 V for pH 4.0 and 5.0, and at around +0.92 V for pH 9.0 and 10.0, which were mainly attributed to the direct oxidation mechanism and the coreactant-mediated oxidation mechanism, respectively. We have supplemented Fig. R4 in Supplementary Fig. 29 in Supporting Information and the related explanation in page 8, lines 4,6.

Figure R4. ECL curves of *t*-COF modified GCEs in (a) acidic and (b) basic PBS containing 20 mM TPrA (PMT voltage = 700 V).

Reviewer #2:

Comment 1: In this work, Lei et al, presented the synthesis of a type of donor-acceptor covalent organic frameworks comprising triphenylamine and triazine units. These COFs were investigated as ECL emitter. The authors tried to establish some correlation between the COF structures and ECL behaviors. Unfortunately, some experimental results and phenomena could be clearly characterized and rationally explained, despite the authors added a large amount of theoretical modeling work. Some fundamental errors could be found. Therefore, we could not recommend such work suitable for submission in the present state. Some issues must be professionally checked and rationally resolved, the detailed was listed below:

Response: We are grateful for the Respected Reviewer's valuable comments. We have improved this work according to your insightful and constructive comments as follows:

Comment 2: The synthetic work is too common to attract the readers within a broad scope. Many similar structures have been reported and fully investigated, although this work tried to open up some special application of COFs.

Response: Thanks a lot for your comments. Indeed, the similar COF structures were reported for the applications in fluorescence sensing, metastructure assembly, and photocatalysis (*J. Am. Chem. Soc.* **2017**, *139*, 8698; *J. Am. Chem. Soc.* **2021**, *143*, 5003; *Angew. Chem. Int. Ed.* **2021**, *60*, 19797). To elucidate the intrareticular charge transfer, we selected this combination of triphenylamine-based COFs as ECL emitters with the different D-A contrast by pairing amino counterparts with different electron affinities. Large D-A contrast of *t*-COF could enable efficient intrareticular charge transfer. What's more, the charge transfer process was further investigated through modulation of crystallinity and protonation of *t*-COF. Therefore, the relationship of COF structure and ECL performance could be deduced during ECL generation, which was rationally confirmed by density functional theory. Finally, an IRCT mediated competitive oxidation mechanism was proposed for these D-A COFs, providing a new fundamental and approach in ECL study. The related discussion was added in page 11, last paragraph, line 1.

Comment 3: The theoretical evaluation seems to occupy a high ratio in this work. As contrast, the experimental data are too less to provide solid supports for the discussion and results. In fact, a lot of characterization and analytic methods have been developed and widely used for the elucidation of semiconducting structure or properties of various organic materials, but not to be used in this work.

Response: Thank you very much for your suggestion. We applied ultraviolet photoelectron spectroscopy (UPS) experiment for elucidating the semiconducting property of three COFs using He I light source (21.22 eV). As shown in Fig. R5a,b,c, the working functions (WFs) of *t*-COF, *b*-COF, and *a*-COF were 3.85 eV, 4.06 eV, and 3.41 eV, respectively. And the VBMs with respect to the Fermi energy of three COFs are 1.36 eV, 1.65 eV, and 1.94 eV. On the other hand, the band gaps extracted from solid-UV data of three COFs are 2.42 eV, 2.53 eV, and 2.28 eV, respectively. Thus the energy levels of three COFs were summarized in Fig. R5d. We have supplemented Fig. R5a,b,c as Supplementary Fig. 10 in Supporting Information and updated Fig. 2b in manuscript. The related explanation was in page 5, last paragraph, lines 3–5.

Figure R5. UPS spectra of (a) *t*-COF, (b) *b*-COF, and (c) *a*-COF. (d) Energy levels of three COFs.

Comment 4: The authors emphasized on the charge transfer between triazine and triphenylamine units, however, the imine linkages between these acceptors and donors seem to seriously impede their communications via electron delocalization. Such interaction is not rational.

Response: Thanks for your valuable comments. In order to verify the charge transfer between triazine and triphenylamine units, we collected the adsorption spectra, photoluminescence (PL) spectra, and time correlated single photon counting (TCSPC) traces of triazine unit (TAPT), triphenylamine unit (TFPA), and *t*-COF. Compared to TAPT and TFPA, the red-shift of adsorption and luminescence indicated the generation of charge transfer state in *t*-COF (Fig. R6a,b) (*Nat. Commun.* **2019**, *10*, 1873). Also, the decrease of τ_1 further proved the charge transfer between

triazine and triphenylamine units in the reticular structure (Fig. R6c and Table R1).

Figure R6. (a) Solid UV-Vis diffuse reflection absorption spectra and (b) PL spectra of TAPT, TFPA, and *t*-COF. (c) TCSPC traces of *t*-COF and TAPT, TFPA in solution and solid state.

Table R1. PL lifetimes of TAPT, TFPA, and *t*-COF.

	τ_1 (ns)	Rel (%)	τ_2 (ns)	Rel (%)
TAPT	0.90	67	2.18	33
TAPT (solid)	0.75	80	3.70	20
TFPA	1.28	25	2.26	75
TFPA (solid)	0.54	94	3.20	6
t -COF	0.49	90	9.66	10

TAPT and TFPA are dissolved in 1,4-dioxane.

Furthermore, we carried out theoretical calculations of the charge distribution of ground state, singlet excited state, and their difference of *t*-COF cell fragment (Fig. R7a,b,c). Upon photoexcitation and electrochemical excitation, the electron density at triphenylamine decreased while that of triazine increased, which indicated the charge transfer process between the donor and acceptor (*Nat. Commun.* **2018**, 9, 3802). The distribution profiles of charge difference between 1st excited and ground state for *t*-COF also agreed with this process (Fig. R7d). In a word, the charge transport could take place across the imine linkages in our conjugated system (*J. Am. Chem. Soc.* **2017**, 139, 8194). We have supplemented Figs. R6,7 and Table R1 in Supplementary Figs. 22,23,37,39,40 and Table S2 in Supporting Information and updated Fig. 3b in manuscript, respectively. The related explanation added in page 7, third paragraph, lines 1–5 and page 9, lines 15–17.

Figure R7. Natural transition orbital (NTO) isodensity surfaces in the (a) ground state and (b) singlet excited state of *t*-COF cell fragment in top and side view. (c) Charge density difference between ground state and singlet excited state of *t*-COF cell fragment. Electron density loss and gain are denoted as blue and purple, respectively. (d) The distribution profiles of charge difference between 1st excited and ground state for *t*-COF. Electron density loss and gain are denoted as red and blue, respectively.

Comment 5: The stability of the resulting COFs against the different pH values had to be examined.

Response: To examine the stability of COFs, we exposed *t*-COF samples to 0.10 M PBS with different pHs for 12 h. We collected the PXRD patterns of these *t*-COF as shown in Fig. R2. The strength of the diffraction peaks at 4.4° indicated that *t*-COFs retained their original skeleton and crystalline structure in pH 6.0–8.0. But at more acidic and basic environment, the crystallinity of *t*-COFs decreased due to hydrolysis. We have supplemented the data in Supplementary Fig. 31 in Supporting Information and the related explanation in page 8, last paragraph, lines 1–2.

Comment 6: Dependence of ECL on crystallinity and corresponding PXRD spectra of *t*-COFs samples from the different ratios of 1,4-dioxane and mesitylene should be further carefully explored and interpreted. Some characterization including fluorescence lifetime measurement etc should be provided.

Response: According to your valuable suggestion, we collected the TCSPC traces of *t*-COFs synthesized with different ratios of 1,4-dioxane and mesitylene (Fig. R8), and the lifetimes were summarized in Table R2. With the increase of crystallinity, τ_1 of *t*-COF decreased from 0.64 ns to 0.54 ns and τ_2 decreased from 6.18 ns to 4.37 ns. The decrease of lifetime indicated the increase of ordered conjugation, which facilitated the intrareticular charge transfer process in *t*-COF and thus accelerated the formation of excited state to enhance ECL. We have supplemented Fig. R8 and Table R2 in Supplementary Fig. 26 and Table S2 in Supporting Information and the related explanation in page 7, third paragraph, lines 9–10.

Figure R8. TCSPC traces of *t*-COFs synthesized with different ratios of 1,4-dioxane and mesitylene of 6:4, 5:5, 4:6, 8:2, and 2:8.

Table R2. PL lifetimes of *t*-COFs synthesized with different ratios of 1,4-dioxane and mesitylene.

	τ_1 (ns)	Rel (%)	τ_2 (ns)	Rel (%)
t -COF (6:4)	0.54	89	4.37	11
t -COF (5:5)	0.61	89	4.73	11
t -COF (4:6)	0.62	89	4.94	11
t -COF (8:2)	0.63	91	5.50	9
t -COF (2:8)	0.64	89	6.18	11

Comment 7: In supplementary Figure 13, the CV profiles look quite bad. It is difficult to find some defined information from the cathodic CV curves of the three COFs. The poor processibility for COF materials might make them not suitable for characterization via CV measurements. The quality of the COF-based electrode should be carefully explored either through processing method or optimizing the additives, e.g. conductive component.

Response: Yes, the conventional COFs are poorly processable for CV measurements. To improve the electrochemical performance, we employed the electroactive ligands to construct three COFs. So, we optimized the cathodic CV tests to obtain the good CV curves. As shown in Fig. R9, the cathodic peaks of *t*-COF, *b*-COF, and *a*-COF were clearly demonstrated at -1.18 V, -1.15 V, and -1.14 V. And the larger current of *t*-COF compared to *b*-COF or *a*-COF was due to the stronger electron affinity of triazine units. We have updated Fig. R9 as Supplementary Fig. 17b in Supporting Information and the related explanation in page 6, third paragraph, line 5.

Figure R9. Cathodic CV curves of *t*-COF, *b*-COF, and *a*-COF modified GCEs in acetonitrile containing 0.10 M TBAPF₆ using Ag/Ag⁺ as reference electrode, and the potential was calibrated to Ag/AgCl.

Comment 8: We could agree with the presented ECL generation pathway of these COF. The resonance structures of aromatic and quinoid types (as shown in Figure 5 or Supplementary Figure 26) seems to be impossible. Either the triazine moiety or imine unit could not undergo such resonance structures due to the large negativity of nitrogen atoms in these units. While, there is no any evidence, such as from ESR or UV-vis-NIR for verifying the proposed mechanism.

Response: According to your suggestions, we collected the adsorption spectra of electrochemically oxidized *t*-COF with different electrolysis time. As shown in Fig. R10a, with the increase of electrolysis time at +1.2 V potential, the adsorption peak appeared at 553 nm increased, which could be attributed to the formed quinoid resonance structure (*J. Mater. Chem. A*, **2019**, 7, 16397; *Angew. Chem. Int. Ed.* **2021**, 60, 12498). Interestingly, following by the reduction under -1.2 V potential or cyclic voltammetry from +1.4 V – -1.4 V, the absorbance of oxidized *t*-COF was restored to initial state, suggesting the transformation between pristine *t*-COF and its quinoid type was relatively reversible (Fig. R10b).

Figure R10. (a) UV-Vis spectra of *t*-COF electrolyzed at +1.2 V with different time. (b) UV-Vis spectra of *t*-COF before (1) and after electrolyzed at -1.2 V for 10 s (2), electrolyzed at +1.2 V for 10 s and followed at -1.2 V for 10 s (3), LSV scan from 0 V – +1.4 V (5), and followed with scan to -1.4 V and then 0 V (4). Scan rate: 0.10 V s⁻¹.

Furthermore, we calculated the charge distribution of one-electron oxidized *t*-COF cell fragment (Fig. R11). Upon removal one electron of the system, there was one electron distributed on NTO 152 and no electron distributed at NTO 153. And the electron on NTO 152 was mainly distributed at the triazine part, suggesting the formation of quinoid species. In summary, both UV-Vis spectra and theoretical calculation confirmed the transformation from aromatic to quinoid resonance structure. We have supplemented Figs. R10–11 as Supplementary Figs. 34,38 in Supporting Information and the related explanation in page 9, lines 5–11.

Figure R11. NTO isodensity surfaces in the oxidized state of *t*-COF cell fragment in top and side view.

Reviewer #3:

Comment 1: *The authors present a novel method of improving the ECL efficiency by controlling the charge transfer processes. A donor-acceptor (D-A) covalent organic framework (COF) with triphenylamine and triazine units is designed as a highly efficient ECL emitter, which could be modulated by crystallinity and protonation. Dual-peaked ECL patterns of COFs are achieved through an IRCT mediated competitive oxidation mechanism: the coreactant-mediated oxidation at lower potential and the direct oxidation at higher potential. This is an interesting work that deserve publication after revision.*

Response: We are very grateful to Reviewer for reviewing the paper so carefully and appreciating the reviewers for acknowledging the strong performance of this work and the good quality of the presentation. We address the comments as follows:

Comment 2: *The bands gaps of the three kinds of COFs were calculated by UV-vis spectra in Figure 2a. The values were 2.42 eV (~513 nm), 2.53 eV (~491 nm), and 2.28 eV (~545 nm) for *t*-COF, *b*-COF, and *a*-COF, respectively. However, the maximum photoluminescence intensities of the three kinds of COFs were observed at ~520 nm for *b*-COF and ~600 nm for *t*-COF and *a*-COF (Figure 2b). I wonder why the stokes shifts were so large. In addition, the bands gaps measured by UV-vis spectra in Figure 2a were larger than the electrochemical results in Figure S13. The cathodic peaks of three COFs were at -1.15 V (Page 6, Line 107) and the anodic peaks of *t*-COF, *b*-COF, and *a*-COF were at 1.15 V, 1.05 V, and 0.85 V (Page 6 , Line 109). As a result, the bands gaps calculated by electrochemical results were 2.3 eV, 2.2 eV, and 2.0 eV for *t*-COF, *b*-COF, and *a*-COF.*

Response: Thanks for your valuable comment. We should note that the UV-Vis spectra of *t*-COF, *b*-COF, and *a*-COF were collected in solid state, and PL spectra were collected in water. As shown

in Fig. R12, the PL peaks of *t*-COF, *b*-COF, and *a*-COF in solid state were 563 nm, 507 nm, and 622 nm while the PL peaks in water were 607 nm, 527 nm, and 622 nm. Besides the energy consumption during relaxation, the large Stokes shift of *t*-COF could originate from the solvent effect, big polarity of water and hydrogen bonds between COF and water (*Angew. Chem. Int. Ed.* **2019**, *58*, 13753).

Figure R12. PL spectra of (a) *t*-COF, (b) *b*-COF, and (c) *a*-COF in the solid state and dispersed in water.

In electrochemical tests, the oxidation involves electron injection from the highest occupied molecular orbital (HOMO) to electrode while the reduction involves electron ejection from electrode to the lowest unoccupied molecular orbital (LUMO). Approximately, the band gap could be regarded as the difference between the oxidation and reduction onset potential. However, the interactions among electrode, solvent molecules, and COFs could affect the CV signal, which could result in the difference between electrochemical and optical band gaps. On the other hand, the interactions between COFs and solvent such as solvatochromism and protonation could result in the red-shift of the UV-Vis spectra (*Nat. Commun.* **2018**, *9*, 3802; *Angew. Chem. Int. Ed.* **2021**, *60*, 19797), and thus the band gaps extracted from the UV spectra in solid state are larger than that in water. Therefore, the optical band gaps calculated by UV-Vis spectra in solid state could be larger than band gaps calculated by electrochemical results in solvent. We added the related explanation in page 6, lines 1,2.

Comment 3: *As demonstrated by the authors, the ECL generation of *t*-COF is very efficient because of the largest donor-acceptor contrast structure of *t*-COF. However, the ECL performance of *t*-COF was only compared with other types of COFs. To further confirm the efficient ECL generation of *t*-COF, the ECL efficiency of *t*-COF, e.g., the relative ECL efficiency of *t*-COF with respect to $\text{Ru}(\text{bpy})_3^{2+}$, should be calculated.*

Response: Thanks a lot for your suggestion. We carried out ECL tests of $1.0 \mu\text{M}$ $\text{Ru}(\text{bpy})_3^{2+}$ and *t*-COF in 0.10 M PBS (pH = 6.8) containing 20 mM TPrA, the relative ECL efficiency of *t*-COF was calculated to be 23.2% with respect to $\text{Ru}(\text{bpy})_3^{2+}$ according to the equation:

$$\Phi_{\text{ECL}} = \Phi_{\text{ECL}}^{\circ} \left(\frac{IQ^{\circ}}{I^{\circ}Q} \right)$$

where Q and Q° are the consumed charges (integrating current vs. time), I and I° are the integrated ECL intensities (integrating ECL spectrum vs. time), and Φ_{ECL} and $\Phi_{\text{ECL}}^{\circ}$ are the ECL

efficiency of the sample and standard, respectively. The $\Phi_{\text{ECL}}^{\circ}$ was set as 1. The relating experiment and the data were added in section of “ECL Imaging” in Supporting Information.

Comment 4: In Figure 2b, the authors compared the photoluminescence intensities of *t*-COF and *a*-COF and the PL intensity of *t*-COF at 607 nm is about 10 times stronger than that of *a*-COF. However, the maximum PL intensities of the two COFs were different. The authors should explain why PL intensity at 607 nm is so important. Moreover, since COF is almost insoluble in all solvents, how to control the two kinds of COF are tested at the same concentration?

Response: The main difference between PL and ECL is that the excited states in PL are generated by photoexcitation while that of ECL are generated by electrochemical reactions. To identify the excited state of ECL emitter, we firstly measured the PL spectrum. In fact, the consistence of ECL spectroscopy (604 nm) with PL spectroscopy (607 nm) of *t*-COF indicated the same excited states. So, we emphasized the PL intensity of *t*-COF at 607 nm.

Indeed, COF is almost insoluble in all solvents. To monitor ECL tests in aqueous solution, we chose water as dispersion solvent. Due to the ordered repetitive structures, the cell weights of *t*-COF, *b*-COF, and *a*-COF could be regarded as their molar weight, which were 629.722 g mol⁻¹, 626.759 g mol⁻¹, and 565.676 g mol⁻¹ (Supplementary Table 5–7), respectively. So all COF dispersions could be controlled to the same concentration based on cell weights of individual COFs. The related description was added in section of “Spectral Measurement” in Supporting Information.

Comment 5: The strong π - π interactions and rotationally labile imine linkages make most of the solid state imine-linked COFs non-fluorescent due to fluorescence quenching processes (Chem. Commun 2018, 54, 2349-2352). For reader's interest, please explain why the three imine COFs in this work have strong fluorescence.

Response: Indeed, most of the solid state imine-linked COFs are non-fluorescent due to the strong π - π interactions and rotationally labile imine linkage. The triphenylamine units of *t*-COF, *b*-COF, and *a*-COF are non-planar building blocks (Fig. R13), which could reduce the π - π interaction. Furthermore, the D-A pairs in *t*-COF enable intrareticular charge transfer, and the fluorescence of *t*-COF could originate from the IRCT state (J. Am. Chem. Soc. 2016, 138, 16703; J. Am. Chem. Soc. 2017, 139, 8194). We have supplemented Fig. R13 as Supplementary Fig. 12 in Supporting Information and the related explanation in page 5, last line and page 6, line 2.

Figure R13. Structure of (a) *t*-COF, (b) *b*-COF, and (c) *a*-COF fragments in top and side views.

Comment 6: In Figure 5e, in basic environment, $t\text{-COF}^{+\bullet}$ was generated from the triphenylamine unit oxidized by either $\text{TPrA}^{+\bullet}$ or electrode. Meanwhile, the triazine unit of $t\text{-COF}$ gained electrons from TPrA^\bullet and generate $t\text{-COF}^{\bullet-}$. Therefore, there are two ECL signals generated from the annihilation between $t\text{-COF}^{+\bullet}$ and $t\text{-COF}^{\bullet-}$ and $t\text{-COF}^{+\bullet}$, respectively. However, in acidic environment, there is only one ECL signal generated from the annihilation between $t\text{-COF}^{+\bullet}$ and $t\text{-COF}^{\bullet-}$. Why different in basic and acid solutions?

Response: Thank you for your valuable comment. We have investigated the dependence of CV of $t\text{-COF}$ and TPrA on pHs. As we can see in Fig. 4b and Fig. R14, the oxidation peaks potentials of $t\text{-COFs}$ kept constant (+1.15 V) while that of TPrA negatively shifted from +1.18 V to +0.70 V with the increase of pHs from 4.0 to 10.0. So in basic solution, TPrA could be oxidized easily and generated $\text{TPrA}^{+\bullet}$, which could react with $t\text{-COF}$ to produce $t\text{-COF}^{+\bullet}$. With the positive scan, the $t\text{-COF}^{+\bullet}$ could be further produced via the direct oxidation at higher potential. Therefore, there were two ECL signals generated from the annihilation between $t\text{-COF}^{+\bullet}$ and $t\text{-COF}^{\bullet-}$. However, in acidic environment, prior to TPrA oxidation, $t\text{-COF}^{+\bullet}$ was obtained via oxidization by electrode, leading to single peaked ECL signal. We have supplemented Fig. R14 as Supplementary Fig. 29a in Supporting Information and added the relating discussion in page 8, line 6.

Figure R14. Dependence of oxidation potentials vs. Ag/AgCl of TPrA on pHs.

Comment 7: In the course of electrode reaction, is $t\text{-COF}$ consumed all the time or recycled like metal complexes, such as $\text{Ru}(\text{bpy})_3^{2+}$?

Response: In our ECL system, $t\text{-COF}$ was oxidized by $\text{TPrA}^{+\bullet}$ or electrode and generated $t\text{-COF}^{+\bullet}$, while $t\text{-COF}$ was reduced by the TPrA^\bullet to obtain $\text{COF}^{\bullet-}$, resulting in the ECL emission via the annihilation between $t\text{-COF}^{+\bullet}$ and $t\text{-COF}^{\bullet-}$ through IRCT. According to the proposed ECL mechanism, the $t\text{-COF}$ could be regenerated. Also, continuous CV scans suggested that $t\text{-COF}$ could recycle like metal complex (Fig. R15a). In addition, the absorbance of $t\text{-COF}$ was nearly recovered after electrolysis at +1.2 V for 10 s and followed by electrolysis at -1.2 V for 10 s (Fig. R15b), which further verified the recyclability of $t\text{-COF}$ during ECL test. We have supplemented Fig. R15b in Supplementary Fig. 34 in Supporting Information and the related explanation in page 9, lines 5–8.

Figure R15. (a) Continuous ECL signals between 0 V and 0.92 V of *t*-COF modified GCE in 0.10 M PBS (pH = 7.6) containing 20 mM TPrA (PMT voltage = 700 V). (b) UV-Vis spectra of *t*-COF before (1) and after (2) electrolyzed at +1.2 V for 10 s and followed with electrolysis at -1.2 V for 10 s (3).

Comment 8: Figure S18 shows the CV curves of *t*-COF, “PMT = 700 V” in the caption is confusing, which might refer to PMT voltage.

Response: Thank you very much for your suggestion. We have revised all the “PMT = 700 V” to “PMT voltage = 700 V”.

Taken together, we wish to express our tremendous gratitude again to the Respected Reviewer for providing this truly insightful suggestion, which has profoundly boosted the significance and novelty of IRCT-mediated COF emitters in ECL fundamentals.

Sincerely yours,

Jianping Lei
Professor of Chemistry

REVIEWERS' COMMENTS

Reviewer #1 (Remarks to the Author):

I am satisfied with the revision and the paper can be published in current form.

Reviewer #2 (Remarks to the Author):

In this updated manuscript, the authors indeed have tried to resolve those concerns from the Reviewers. Combining the newly added characterization and discussion with the original description, I think that the current version could be considered for publication in Nature Communication.

Reviewer #3 (Remarks to the Author):

This manuscript presented a methodology of improving the ECL efficiency by controlling the charge transfer processes in a donor-acceptor covalent organic framework (COF) with triphenylamine and triazine units. Moreover, dual ECL emissions were achieved at low and high potentials. Thus a system holds promise for both sensitive and multiplexed analysis.

The authors have fully considered all my comments to prepare the revised manuscript that I believe deserve publication by the journal.

Dr. Prof. Jianping Lei
State Key Laboratory of Analytical Chemistry for Life Science
School of Chemistry and Chemical Engineering
Nanjing University
Nanjing 210023, PR China
Email: jpl@nju.edu.cn

October 20, 2021

Re: “Intrareticular charge transfer regulated electrochemiluminescence of donor-acceptor covalent organic frameworks” (NCOMMS-21-19890A)

Reviewer #1:

Comment 1: I am satisfied with the revision and the paper can be published in current form.

Response: We thank the reviewer for this positive comment and support the publication of our manuscript.

Reviewer #2:

Comment 1: In this updated manuscript, the authors indeed have tried to resolve those concerns from the Reviewers. Combining the newly added characterization and discussion with the original description, I think that the current version could be considered for publication in Nature Communication.

Response: We thank the reviewer for the constructive comments and suggestions, which have made our manuscript much improved.

Reviewer #3:

Comment 1: This manuscript presented a methodology of improving the ECL efficiency by controlling the charge transfer processes in a donor-acceptor covalent organic framework (COF) with triphenylamine and triazine units. Moreover, dual ECL emissions were achieved at low and high potentials. Thus a system holds promise for both sensitive and multiplexed analysis.

The authors have fully considered all my comments to prepare the revised manuscript that I believe deserve publication by the journal.

Response: We thank the reviewer for this positive comment and support the publication of our manuscript.

Thank you again for the reviewers' consideration of our manuscript.

Sincerely yours,

Jianping Lei
Professor of Chemistry